# Relationship between Down-Regulation of Copper-Related Genes and Decreased Ferroportin Protein Level in the Duodenum of Iron-Deficient Piglets

**DOI:** 10.3390/nu13010104

**Published:** 2020-12-30

**Authors:** Aneta Jończy, Rafał Mazgaj, Rafał Radosław Starzyński, Piotr Poznański, Mateusz Szudzik, Ewa Smuda, Marian Kamyczek, Paweł Lipiński

**Affiliations:** 1Department of Molecular Biology, Institute of Genetics and Animal Biotechnology, PAS, 05-552 Jastrzębiec, Poland; r.mazgaj@igbzpan.pl (R.M.); r.starzynski@igbzpan.pl (R.R.S.); m.szudzik@igbzpan.pl (M.S.); e.smuda@igbzpan.pl (E.S.); 2Department of Experimental Genomics, Institute of Genetics and Animal Biotechnology, PAS, 05-552 Jastrzębiec, Poland; p.poznanski@igbzpan.pl; 3Pig Hybridization Centre, National Research Institute of Animal Production, 64-122 Pawłowice, Poland; marikamy@wp.pl

**Keywords:** anemia, copper, duodenum, iron-deficiency

## Abstract

In mammals, 2 × 10^12^ red blood cells (RBCs) are produced every day in the bone marrow to ensure a constant supply of iron to maintain effective erythropoiesis. Impaired iron absorption in the duodenum and inefficient iron reutilization from senescent RBCs by macrophages contribute to the development of anemia. Ferroportin (Fpn), the only known cellular iron exporter, as well as hephaestin (Heph) and ceruloplasmin, two copper-dependent ferroxidases involved in the above-mentioned processes, are key elements of the interaction between copper and iron metabolisms. Crosslinks between these metals have been known for many years, but metabolic effects of one on the other have not been elucidated to date. Neonatal iron deficiency anemia in piglets provides an interesting model for studying this interplay. In duodenal enterocytes of young anemic piglets, we identified iron deposits and demonstrated increased expression of ferritin with a concomitant decline in both Fpn and Heph expression. We postulated that the underlying mechanism involves changes in copper distribution within enterocytes as a result of decreased expression of the copper transporter—Atp7b. Obtained results strongly suggest that regulation of iron absorption within enterocytes is based on the interaction between proteins of copper and iron metabolisms and outcompetes systemic regulation.

## 1. Introduction

Iron and copper belong to the group of trace elements required for physiological functions and development of the body. In biological systems, they exist in two oxidation states, which determine their main role—acting as cofactors of numerous cellular enzymes. On the other hand, intracellular iron and copper associated with a variety of ligands participate in the generation of powerful oxidants, which directly damage proteins, lipids, and nucleic acids [1]. Therefore, metabolisms of both metals must be tightly controlled at the systemic and cellular levels. The metabolic links between iron and copper have been recognized for many years [2]; however, detailed molecular mechanisms of interactions are still unknown. During iron deficiency, copper levels are significantly increased in the liver and serum [3,4], while copper deficiency leads to the hepatic and intestinal accumulation of iron, and thus decreases iron availability for erythropoiesis and contribute to the development of anemia in rodents, pigs, and humans [5,6,7,8]. According to the World Health Organization, anemia is one of the most common and widespread disorder worldwide [9]. Among livestock, iron deficiency anemia (IDA) occurs most often in pigs and is the most frequent deficiency disorder during the early postnatal period [10]. Therefore, iron supplementation of suckling piglets is a common practice in the swine industry and efficiently counteracts the development of IDA [11]. In addition, early studies reported severe hypochromic and microcytic anemia associated with dietary copper deficiency in pigs [6,12]. It seems that intersecting copper and molecular iron pathways in duodenal enterocytes are largely responsible for the assimilation of both microelements from the diet [13]. Absorption of both metals requires the reduction of Cu^2+^ and Fe^3+^ ions on the surface of enterocytes, which involves duodenum cytochrome b_561_ reductase (Dcytb), located in the apical membrane of enterocytes [14,15]. After reduction, Cu^+^ and Fe^2+^ ions are transported into enterocytes by copper transporter 1 (Ctr1) and divalent metal transporter 1 (Dmt1), respectively [16,17]. In addition, Dmt1 is thought to be responsible for copper uptake during IDA in mice [18]. Depending on the systemic demand, iron taken up from the diet can be stored in the complex with intra-enterocytic ferritin (Ft) [19] or transported through the basolateral membrane of enterocytes into the blood by the only known mammalian iron exporter—ferroportin (Fpn) [20,21]. Effective iron export by Fpn requires the activity of copper-dependent, membrane-anchored ferroxidase called hephaestin (Heph). Mutations in the gene encoding Heph lead to iron retention in enterocytes and, in consequence, to systemic anemia [22]. It has been postulated that Atp7b, a copper-transporting ATPase, is involved in the incorporation of Cu ions into apo-Heph [23], although direct evidence of this process is still lacking.

Systemic regulation of iron absorption involves hepcidin, a small peptide hormone produced by hepatocytes that adjusts dietary iron intake to body iron requirements. Hepcidin binds to Fpn to induce its degradation, thus inhibiting iron export from enterocytes [24]. Under certain physiologic and pathologic conditions requiring increased iron supply such as iron deficiency, enhanced erythropoiesis, hypoxia, and growth, hepcidin expression is suppressed [25]. In addition to systemic factors, local intestine-specific regulation of iron handling seems to be crucial for maintaining body iron homeostasis [26], which is especially important during the early stage of postnatal development. Iron deficiency and hypoxia potentiate the expression of hypoxia-inducible factor-2 alpha (Hif-2α) in the intestine, which transcriptionally activates a set of genes involved in iron absorption [27,28,29]. Hif-2α increases the transcription of copper-transporting ATPase—Atp7a, which is an enterocyte copper exporter [30], moreover Hif-2α controls the constitutive expression of duodenal copper transporter Ctr1 [31].

Considering numerous physiological interplays between copper and iron, this study attempted to investigate the influence of copper-related genes on iron absorption in the duodenum of animals with different iron statuses. For this purpose, we used an animal model of suckling piglets with naturally occurring IDA and iron-replete animals subjected to iron supplementation therapy. Our results suggest that in anemic piglets, intra-enterocytic regulation of iron absorption is largely based on the interaction between copper and iron metabolism proteins and dominates systemic control of iron influx to the body across the duodenum.

## 2. Materials and Methods

### 2.1. Animals

The experiment was conducted at the Pig Hybridization Centre in Pawłowice (NRIAP, National Research Institute of Animal Production, Balice, Poland). A total of 27 male piglets (line 990) from different litters were allotted to 3 experimental groups: three-day-old piglets; 28-day-old piglets intramuscularly injected with 80 mg Fe/kg b.w in the form of iron dextran, FeDex (Ferran 100, Vet-Agro, Lublin, Poland) on day three after birth; 28-day-old piglets intramuscularly injected with 0.9% NaCl on day three after birth. The group of three-day-old piglets was used as a starting point to elucidate the progression of anemia in piglets deprived of iron supplementation. Based on the previous research and obtained body mass of the piglets, 80 mg Fe/kg b.w in the form of iron dextran was injected [32,33]. Piglets were fed the Prestarter Wigor 1 Plus feed (manufactured at the feed mill of the Experimental Station of the NRIAP and used as a standard type of diet in nursery piglets) containing 150 mg Fe and 160 mg Cu per kg of dry feed and housed in standard conditions (70% humidity and a temperature of 22 °C in cages with straw bedding). During the 28-day experiment, sows were allowed to nurse their piglets. Piglets were euthanized by intracardiac injection of a high dose of sodium pentobarbital (Morbital, Biowet, Puławy). All experimental protocols were approved by the Second Local Ethical Committee on Animal Testing in Warsaw (permission no. 26/2017).

The 6-month-old mutant C3HeB/FeJAtp7btx-J/J and control C3HeB/FeJ mice were kindly provided by Dr Małgorzata Lenartowicz (Jagiellonian University, Cracow, Poland). Mice were maintained under a 12-h light/12-h dark cycle (lights on at 7 a.m.) and had ad libitum access to standard murine chow (LABOFEED H, Kcynia, Poland) and tap water.

### 2.2. Biological Sample Collection

Blood samples were collected into heparin-coated tubes, directly from the heart of piglets after euthanasia, and centrifuged (400× *g*, 5 min, room temperature RT) to separate the plasma. Plasma samples were aliquoted and stored at −80 °C until use. The fragments of liver, spleen, and the brain of piglets were washed in PBS, frozen in liquid nitrogen for molecular analyses, or fixed in Bouin’s solution for iron staining analysis. The proximal duodenal segment was rinsed with PBS and cut into equal parts. One part was used for iron staining analyses, the other part was further dissected, and then a scalpel was used to scrape the upper layer of the duodenum to obtain the fraction enriched in duodenal epithelial cells (enterocytes). Duodenal scrapings were stored for further analysis at −80 °C. Piglet femurs were mechanically cleansed of the surrounding muscles; the bone marrow was washed out with PBS using a syringe. The bone marrow suspension was centrifuged (400× *g*, 5 min, room temperature (RT)), and the cell pellet was frozen and stored at −80 °C. Hematological indices were measured using the veterinary Scil Vet ABC analyzer (Horiba, Japan).

Control and C3HeB/FeJAtp7btx-J/J mutant mice were euthanized by cervical dislocation; the duodenum of mice were rinsed with PBS and fixed in Bouin’s solution for iron staining analysis.

### 2.3. Measurement of Non-Heme and Heme Iron Content in the Liver

Non-heme iron content in the liver of piglets was determined using colorimetric assay as described previously [34]. Briefly, liver fragments were homogenized in PBS followed by digestion in 25% trichloroacetic acid solution for 10 min at 100 °C. The mixture was then centrifuged (9200× *g*, 5 min, RT). The supernatant was transferred to new tubes containing the reaction buffer (3 M sodium acetate, 230 mM sodium ascorbate, and 10 mM ferrozine). The absorbance was measured at 562 nm (Beckman DU-68, Beckman Coulter, Brea, CA, USA). To determine the heme iron content in the liver, tissue was homogenized in 85% formic acid and incubated for 15 min at RT. The suspension was centrifuged (15,600× *g*, 2 min, RT), and 50 µL of the supernatant was transferred back into 450 µL formic acid. Absorbance was measured at 400 nm (Beckman DU-68).

### 2.4. Measurement of Copper Content in the Tissues

Copper concentration was determined in the liver, spleen, and brain of piglets by atomic absorption spectroscopy. Samples were digested in 2 mL of boiling Suprapur-grade nitric acid (Merck, Darmstadt, Germany). After cooling to RT, each sample was suspended in 10 mL of deionized water. Reference material samples were prepared in a similar manner. The copper concentration was measured using the graphite furnace atomic absorption spectrometry (AAS) technique (AAnalyst 800, Perkin-Elmer, Waltham, MA, USA).

### 2.5. Perls Staining

Microscopic slides were dipped five times in gelatin-coating solution, obtained by dissolving 5 g gelatin and 0.5 g chromium potassium sulfate dodecahydrate in 1 L of deionized water. Then, slides were left to dry at RT for 48 h in a dust-free environment and stored for further use. The tissues fixed in Bouin’s solution were washed with 70% ethanol and stored for further processing. After dehydration, tissues were embedded in paraffin and cut into 10 µm sections on a microtome (Hyrax M25, Zeiss), placed on gelatin-coated microscopic slides, and air-dried at RT. To localize iron deposition, tissue iron staining (Perls staining) was performed. Briefly, tissue sections were deparaffinized in xylene (2 × 5 min) and hydrated to distilled water. Then samples were incubated for 45 min in staining solution, obtained by mixing equal parts of 20% aqueous hydrochloric acid and 10% aqueous potassium ferrocyanide. After that, slides were washed in three changes of distilled water and counterstained by 5 min incubation in 0.02% pararosaniline solution. Subsequently, samples were rinsed in distilled water, dehydrated in 95% and 99.8% ethanol (2 × 1 min), cleared in xylene (2 × 3 min), and coverslipped with DPX (Sigma-Aldrich, St. Louis, MO, USA). Samples were analyzed under standard light microscopy (Eclipse E200, Nikon).

### 2.6. Quantitative Reverse Transcription PCR (RT-qPCR)

Total RNA from the liver and duodenum of piglets was isolated using the High Pure RNA Tissue Kit (Sigma Aldrich, St. Louis, MO, USA) according to the manufacturer’s protocol. Total RNA from bone marrow cell pellet was extracted with Trizol Reagent (Thermo Fisher Scientific, Waltham, MA, USA). One microgram of total DNAse-treated RNA of liver and duodenum samples and 700 ng of RNA from bone marrow cells were reverse transcribed using a Transcriptor First Strand cDNA Synthesis Kit (Roche Diagnostics, Mannheim, Germany). The amplified products were detected using SYBR Green I (Roche Diagnostics, Mannheim, Germany). Real-time quantitative PCR analysis was performed in a Light Cycler U96 (Roche Diagnostics, Mannheim, Germany) using gene-specific primer pairs (Appendix A). Transcript levels were normalized relative to 18S rRNA, succinate dehydrogenase complex, subunit A (Sdha), hypoxanthine-guanine phosphoribosyltransferase (Hprt), and TATA-binding protein (Tbp) expression levels. Control reference genes have been selected using NomFinder software (v0.953, https://moma.dk/normfinder-software). Analysis of RT-qPCR results based on -ΔCt (cycle threshold) (Ct of reference gene—Ct of test gene).

### 2.7. Protein Extracts Preparation and Western Blotting

For the analysis of proteins crude membrane and cytosolic protein, extracts were prepared from the liver and duodenal scrapings as described previously [35]. Briefly, tissues were homogenized in sucrose histidine buffer (0.25 mol/L sucrose, 0.03 mol/L histidine, pH 7.2), supplemented with a cocktail of protease inhibitors (Sigma-Aldrich, St. Louis, MO, USA). The homogenates were centrifuged (6000× *g*, 15 min, 4 °C) followed by ultracentrifugation of supernatant (80,000× *g*, 45 min, 4 °C). The supernatants obtained after ultracentrifugation and enriched in cytosolic proteins fraction were collected and stored at −80 °C. Final pellets were resuspended in sucrose histidine buffer and stored at −80 °C until use. Protein concentrations were determined by the Bradford assay (Biorad, Hercules, CA, USA), and protein extracts were subjected to SDS-PAGE electrophoresis. Before membrane blocking with 5% skimmed milk, transfer efficiency was confirmed by Ponceau-S staining. A list of primary and secondary antibodies used is shown in Appendix A. Validation of the cross-reactivity of primary polyclonal antibodies raised against mouse Fpn and LFt has been done in previous research [32,36]. Based on the specificity provided by the commercial suppliers, the primary anti-Smad4 antibody recognizes pig protein. Additionally, we performed Blastp analysis for analyzed proteins, and results are shown in Appendix A. After incubation with primary antibody at 4 °C overnight, the blots were extensively washed and incubated with the horseradish peroxidase-conjugated secondary antibodies at RT for 1 h followed by visualization with the enhanced luminescence kit (Advansta, Menlo Park, CA, USA).

### 2.8. Superoxide Dismutase 1 (Sod1) and Ceruloplasmin (Cp) Activity Assay

The cytosolic protein extract obtained from duodenal scrapings was subjected to protein electrophoresis under non-reducing and non-denaturing conditions. The separation was carried out at 4 °C and a constant voltage of 160 V for 2 h. After electrophoresis, the gel was transferred to reaction buffer (0.13 mg/mL nitro blue tetrazolium chloride; 0.1 mg/mL riboflavin; 1 μL/mL TEMED; 50 mM KPO_4_ solution (pH 7.8)) and incubated for 45 min in the dark at RT. After the incubation, the gel was irradiated (visible light) until the visible bands appeared in the gel. Control of the amount of Sod1 protein in the sample loaded on the gel was estimated based on the Sod1 protein level calculated after densitometric analysis of Sod1 versus β-actin (reference protein) bands visualized following SDS-PAGE electrophoresis of the same cytosolic extract. Ceruloplasmin activity in the serum was determined using a colorimetric assay kit (Thermo Fisher Scientific, Waltham, MA, USA) according to the manufacturer’s protocol.

### 2.9. Statistical Analysis

Data are presented as mean values ± standard deviation (SD). Statistical analysis, which included results obtained in the groups of three- and 28-day-old piglets, was performed using a two-way ANOVA, applying a test of outliers elimination (ROUT, Robust regression and outlier removal (Q = 1%)) and checking that the data are normally distributed (Shapiro-Wilk test). Dunnet’s multiple comparison post-hoc test was used. Statistical analysis within two groups of 28-day-old piglets, characterized by varied iron status, was performed using Student’s *t*-test for independent groups. Statistical analysis was performed using GraphPad Prism 8.4.2 software (GraphPad, San Diego, CA, USA). *p* ≤ 0.05, *p* ≤ 0.01, and *p* ≤ 0.001 were considered significant and are denoted with one, two, or three asterisks, respectively.

## 3. Results

### 3.1. Red Blood Cell (RBC) Parameters and Iron Status of Piglets During an Early Stage of Postnatal Development Indicate Iron Deficiency Anemia

Neonatal IDA is a common pathology that affects piglets of all breeds [37]. In 28-day-old piglets deprived of iron supplementation, a significant decrease in the values of most RBC indices such as hemoglobin (HB) concentration, hematocrit (HCT) level, and the mean corpuscular volume (MCV) was observed (Table 1). However, there were no changes in the erythrocyte count compared to the group of three-day-old individuals. Piglets receiving a high amount of FeDex by intramuscular injection showed significantly higher values of RBC indices (Table 1).

The varied RBC status observed in the groups of 28-day-old piglets was also reflected in the transcriptional activation of the gene encoding erythropoietin (Epo) and erythroferrone (Erfe), synthesized in the kidney and bone marrow during stress erythropoiesis, respectively [38,39]. Both Epo and Erfe expression in the tissues of anemic piglets has been shown to be higher than that observed in FeDex-supplemented animals (Figure 1A). These data confirm that without iron supplementation, suckling piglets develop systemic iron deficiency anemia. A significant increase in non-heme and heme iron content in the liver of 28-day-old piglets injected with FeDex was noted (Figure 1B). Importantly, intramuscular FeDex injections led to the preferential iron accumulation in the liver macrophages (Browicz–Kupffer cells) (Figure 1C), but no iron deposits were observed in hepatocytes of FeDex-supplemented piglets.

### 3.2. Increased Copper Levels in Organs Controlling Iron Metabolism but Not in the Duodenum of Anemic Piglets

Analysis of the total copper content in the liver and spleen of anemic piglets showed a statistically significant increase, compared to the level found in the organs of pigs supplemented with FeDex (Figure 2A). Among all groups, no significant changes were observed in the brain—an organ not directly involved in regulating iron metabolism (Figure 2A). Despite increased hepatic copper concentration, no changes in biliary copper content were observed in the anemic piglets, although the expression of Atp7b responsible for copper excretion into the bile decreased significantly (Figure 2B). Moreover, there were no changes in plasmatic ceruloplasmin (a copper-dependent enzyme or ferroxidase) expression or activity (Figure 2C). In order to determine the status of copper in enterocytes, the copper chaperone for superoxide dismutase (Ccs) and superoxide dismutase 1 (Sod1) protein levels and Sod1 activity was examined using cytoplasmic extracts of the intestinal scrapings. However, in our study, no differences were found neither in Ccs protein level nor in the activity of Sod1 between studied groups (Figure 2D).

### 3.3. Iron Accumulation and Suppressed Hif-2α Transcriptional Activity in the Duodenum of Anemic Piglets

Despite severe systemic anemia in piglets deprived of FeDex supplementation, duodenal ferritin (iron storage protein) abundance was increased, and Perls staining revealed iron deposits localized in enterocytes of anemic piglets (Figure 3A). It is important to note that the sensitivity of the Perls stain to detect low amounts of iron is limited, therefore observed iron deposits confirm increased iron storage in the duodenum of anemic piglets. Interestingly, in the duodenum of anemic piglets, increased Smad4 protein level was observed (Figure 3B). Smad4 is a transcription factor upregulated by iron that limits Hif-2α transcriptional activity [40]. To investigate whether increased Smad4 level and iron deposition in the duodenum of anemic piglets could influence Hif-2α–dependent transcriptional response of genes responsible for iron absorption during anemia, we checked mRNA expression of particular genes. Interestingly, expression levels of the HIF-2α–responsive iron-related genes Cybrd1 and Dmt1 as well as copper transporter Atp7a were not significantly elevated in the duodenum of anemic piglets compared to FeDex-supplemented animals. Moreover, Ctr1 mRNA level, which basal expression in intestinal epithelial cells is also regulated by Hif-2α [31], was significantly decreased (Figure 3C).

### 3.4. Decreased Expression of Copper Transporter Atp7b Contributes to Iron Retention in the Duodenum of Anemic Piglets

To understand the mechanism responsible for iron retention in the duodenum of anemic piglets, we investigated the expression of Fpn, a sole cellular iron exporter. We found that Fpn protein levels in the membrane fraction of extracts obtained from the duodenum of anemic piglets was significantly lower than in the group of pigs supplemented with FeDex (Figure 4A). The Fpn protein level was comparable to that observed in the group of 3-day-old piglets, in which mechanisms of iron absorption are not yet developed. The decrease in Fpn protein expression was not accompanied by a decrease in Fpn mRNA expression, suggesting post-translational hepcidin-dependent regulation. The level of Fpn expression may be the result of the interaction of this protein with the superior regulator of its expression—hepcidin, produced mainly in the liver [41]. However, despite significant differences in the liver iron content of anemic and FeDex-supplemented piglets, the level of hepcidin expression did not differ within the studied groups (Figure 4B). Fpn is able to form a functional complex with a copper-dependent, membrane-like protein that possesses ferroxidase activity—hephaestin (Heph) [42]. We found that in the intestine of anemic piglets, the level of Heph expression, both at the mRNA and protein level, was significantly lower compared to FeDex-supplemented piglets (Figure 4C). Heph stability in the intestine is presumed to be dependent on the activity of the Atp7b protein [23]. Importantly, Atp7b mRNA expression in the intestine of anemic piglets was significantly lower compared to that found in the intestine of FeDex-supplemented piglets (Figure 4D). Moreover, in the duodenum of C3HeB/FeJAtp7btx-J/J mutant mice carrying the mutation of Atp7b gene and characterized by nonfunctional Atp7b protein, iron deposits were localized (Figure 4E).

## 4. Discussion

The interactions between copper and iron metabolisms have been known for many years; however, elementary questions have not yet been answered. Why does the level of copper in the liver increase significantly during anemia, and what is the molecular basis of this phenomenon? Does iron mediate these processes? In turn, in the case of copper deficiency, a major question arises as to the molecular mechanisms of iron accumulation in the duodenum, liver, and spleen, which may contribute to the development of anemia. In the present study, based on the model of neonatal IDA developing naturally in piglets, an attempt was made to determine the molecular interaction of copper and iron metabolisms, with particular emphasis on the processes associated with the absorption of both elements.

The etiological factors contributing to the rapid development of neonatal anemia in piglets include low levels of iron accumulated in the liver during pregnancy (one of the lowest among the all mammalian species), low iron content in colostrum and sow’s milk [10,11], as well as poorly developed mechanisms of iron absorption in the intestine of piglets [32]. A general consensus regarding hemoglobin concentration sets the cut-off level of anemia in suckling piglets at 8 g/dL [43]. The three- and 28-day-old piglets not supplemented with FeDex used in this study showed hemoglobin values largely below this threshold. During anemia, an increased iron requirement for hemoglobin production leads to the activation of systemic mechanisms that stimulate the absorption of this micronutrient in the duodenum. One is the synthesis of erythropoietin (Epo) in the kidneys [44]. Epo, by binding to its receptors in erythrocyte progenitor cells, stimulates their proliferation and differentiation [45]. Epo also stimulates the synthesis of erythroferrone (Erfe)—a hormone produced in the bone marrow by erythrocyte precursor cells, responsible for the suppression of hepcidin expression in the liver [39], thereby contributing to increased iron absorption in the duodenum and iron supply for the erythropoiesis. In this context, the observed increase in the expression of *Epo* and *Erfe* genes in the kidney and bone marrow of 28-day old non-supplemented piglets further confirms the anemic status and enhanced systemic iron demand of these animals.

Supplementation of piglets with iron between days one and four after birth is a routine veterinary practice, which aims to supply exogenous iron for current needs related to the intensive course of erythropoiesis. In accordance with previous studies [46,47], intramuscular injection of FeDex significantly improved the hematological indices in piglets, preventing the development of anemia. Moreover, iron content in the liver of FeDex-supplemented piglets increased, and iron deposits were localized within the liver macrophages as previously described [32]. This emphasizes the specific pattern of iron distribution in the liver after iron dextran (FeDex) supplementation and attests the iron-replete status of animals [48].

The hepatic pattern of copper content and redistribution in iron-deficient and iron-replete piglets largely confirms previous results reported in rodents [4,49]. An increase in copper content was observed, both in the liver and spleen of anemic piglets, organs directly involved in the systemic regulation of iron homeostasis. Some research suggested that besides potentially increased copper absorption during iron deprivation in rats, disturbances in the excretion of copper into the bile mediated by Atp7b protein could be responsible for increased hepatic copper storage [4]. However, they noticed that hepatic Atp7b gene expression was not altered during iron deficiency. In contrast, the present data show that the level of expression of this gene in the liver is significantly decreased in anemic piglets but without a concomitant decrease in biliary copper excretion. In the liver, Atp7b also mediates the process of Cu ions incorporation into ceruloplasmin (Cp) molecule, the main copper-dependent plasma ferroxidase, transforming enzymatically inactive and unstable apo-Cp into the active enzyme—holo-Cp [50,51,52]. The role of Cp in maintaining iron homeostasis has been made evident in iron overloaded phenotype in humans that lack Cp [53]. Previous research revealed increased Cp expression and activity in the serum of anemic rats, suggesting that Cp induction would increase iron mobilization from the storage compartments [54]. However, in our study, we did not observe any differences in the plasma Cp level or activity.

Regarding duodenal copper status in iron insufficiency, it has been proven in studies on iron-deficient rats that copper accumulates in enterocytes [4]. Meanwhile, our evaluation of Ccs and Sod1 protein levels, as well as cytoplasmic Sod1 activity in the duodenum of iron-deficient piglets, did not reflect differences in copper status between iron-deficient and iron-replete piglets. The Ccs protein is considered a marker of the copper content in the cells and is responsible for transport and delivery of Cu ions to superoxide dismutase 1 (Sod1, Cu, Zn-Sod1)—a copper-dependent enzyme possessing antioxidant activity [55]. The steady-state Ccs levels are inversely proportional to intracellular Cu availability, and under copper-deficient conditions, the level of Ccs protein (but not mRNA) increases, while Sod1 activity decreases [56,57]. Our results indicate that in the intestine of piglets, regardless of their systemic iron status, a certain stable, bioavailable pool of copper is maintained, which to some extent can protect the cell from the potentially negative effects of overload or deficiency of this element, and thus change the activity of copper-dependent enzymes.

Decreased RBC and hemoglobin content during iron deficiency anemia leads to systemic tissue hypoxia [58]. The HIF transcription factors are central mediators of cellular adaptation to low oxygen (O_2_) tension. Importantly, it has been shown that Hif-2α, a regulatory HIF subunit, is stabilized when O_2_ is scarce or under iron-deficient conditions [59]. Intestinal iron absorption requires Hif-2α as a main positive transcriptional regulator of genes responsible in the duodenum for both iron uptake from the diet (Dmt1, Cybrd1) [27,29] and iron export into the bloodstream (Fpn) [28]. In mice, the intestinal epithelial iron level seems to be the main factor that controls Hif-2α activity by regulating iron-dependent intestinal prolyl hydroxylase domain enzymes [26]. Therefore, even upon increased systemic iron demand, the transcriptional Hif-2α response can be reduced in the iron-rich duodenum. It has been proven in both in vitro and in vivo conditions that Smad4, an iron-dependent transcription factor, can limit Hif-2α transcriptional activity and iron deficiency leads to proteasomal degradation of Smad4 [42]. In the small intestine of mice fed iron-enriched, iron-replete, or low-iron diets, a dose-dependent decrease in Smad4 level has been demonstrated. In the duodenum of anemic piglets, despite systemic anemia, increased ferritin expression and iron deposition within duodenal villi clearly denote iron accumulation, leading to the increased Smad4 expression, which in turn may result in reducing Hif-2α transcriptional activity. These data indicate that intestinal epithelial iron levels may decrease trans and/or cis-acting Hif-2α activity during systemic iron deficiency anemia in piglets. Therefore, in the duodenum of anemic piglets, we did not observe significantly increased expression, neither of iron-related (Cybrd1 and Dmt1) nor copper-related genes (Atp7a and Ctr1)—all upregulated by Hif-2α.

Increased expression of proteins involved in the transfer of iron across the apical membrane of enterocytes could be responsible for the accumulation of iron in intestinal villi observed in iron-deficient piglets. However, no differences were found in the expression of both Cybrd1 and Dmt1 genes at the mRNA level in the duodenum of iron-replete and iron-deficient animals. This observation allows us to conclude that the process of iron transport into the enterocytes is not a limiting factor responsible for the increased content of this microelement in the duodenum of anemic piglets. We, therefore, tested the possible contribution of reduced iron egress from the enterocytes as a cause of iron retention in these cells. Ferrous ions (Fe^2+^) are exported from enterocytes by ferroportin (Fpn), the only known vertebrate iron exporter [20,21,32]. In the intestine, Fpn is located in the basolateral membrane of mature enterocytes [60,61]. The conditional intestine-specific knockout of the *Fpn* gene, quickly results in iron retention within enterocytes and the development of iron deficiency anemia in mice [60]. Iron deficiency induces Hif-2α binding to the HRE (hypoxia-response element) sequence in the promoter region of the Fpn gene, leading to an increase at the level of mRNA [28]. However, in the duodenum of anemic piglets, no change of Fpn transcript abundance was observed compared to iron-replete piglets. This excludes the participation of Hif-2α in the regulation of Fpn expression in enterocytes of these animals. Interestingly, in the duodenum of anemic piglets, a significantly lower Fpn protein level was observed compared to piglets supplemented with FeDex. The master regulator of Fpn expression at the protein level is hepcidin, a peptide mainly produced by hepatocytes in response to increased iron levels [43]. Hepcidin, by binding to Fpn, stimulates its endocytosis and degradation [24]. Analysis of the expression of the Hamp gene (coding for hepcidin) in the liver showed no difference between anemic and FeDex supplemented piglets, despite the high levels of iron accumulated in the liver of the latter. This may be related to the specific distribution of iron administered to piglets in the form of FeDex, indicating the deposition of iron within the liver macrophages, not endothelial cells, which are primarily responsible for controlling the iron expression of hepcidin [62]. Taking into consideration the findings mentioned above, it is clear that extracellular hepcidin-dependent regulation of Fpn expression is not responsible for its down-regulation in the duodenum of iron-deficient piglets. Interestingly, previous research has indicated that even an increased level of hepcidin expression did not have an influence on Fpn protein abundance in the duodenum of young piglets [32], which could suggest that systemic regulation of Fpn is immature in newborn piglets or Fpn is poorly sensitive to hepcidin.

Looking for a possible cause of varied Fpn protein levels in the duodenum of piglets, we focused our investigations on the interaction between Fpn and hephaestin (Heph), a copper-dependent ferrous iron oxidase [63]. Fpn acts in conjunction with the intestinal Heph to mediate iron release from the enterocyte. Heph has been shown to co-localize with Fpn within the cell membrane, forming a complex that allows Fe ions to be exported to the extracellular environment [44]. Of note, significant iron retention within enterocytes has been demonstrated in the duodenum of Heph knockout mice [64]. In the intestine of anemic piglets, a decrease in Heph levels was demonstrated at both the mRNA and protein levels. It has been suggested that the expression of the *Heph* gene could be predominantly responsive to systemic levels of iron and are independent of iron levels in the enterocytes [65]. The transformation of apo-Heph into the enzymatically active form of holo-Heph, resulting from the incorporation of Cu ions into a protein molecule, reduces the rate of its degradation in proteasomes [66].

As mentioned before, it is postulated that the Atp7b protein is involved in the incorporation of Cu ions into apo-Heph. Interestingly, the knockout of the Atp7b gene in mice, in addition to the liver dysfunction typical for this genetic modification, also leads to iron accumulation in the duodenum [23]. The role of Atp7b protein in the intestine for a long time was marginalized, while most analyzes were focused on Atp7a, which is the main exporter of Cu ions from enterocytes. Atp7b protein has a slightly lower affinity for Cu ions than Atp7a; however, the rate of catalytic processes of phosphorylation and dephosphorylation of the Atp7a protein (which determines the efficiency of Cu ion transport) is about six times higher compared to Atp7b [67]. These data suggest that Atp7a is able to transport Cu ions much faster and more efficiently, and probably that is why it is involved in the export of Cu ions to the bloodstream. The Atp7b protein does not participate in the process of copper export from enterocyte and may be responsible for other functions in the duodenum. Histologic staining demonstrated significant Fe accumulation in the duodenum of Atp7b knockout mice [23]. Using C3HeB/FeJAtp7btx-J/J mouse model carrying the mutation of Atp7b gene, we confirmed iron deposition in the duodenum of mice characterized by nonfunctional Atp7b protein. Interestingly, in the intestine of anemic piglets, despite severe systemic anemia, iron deposits and a concomitant decrease in Atp7b transporter expression was noted. This suggests that the decrease in Heph levels in the duodenum of anemic piglets may be the result of a decreased Atp7b expression, which could be responsible for the defective process of embedding Cu ions into the structure of the unstable form of apo-Heph.

## 5. Conclusions

In conclusion, a potential mechanism responsible for the observed increase in iron levels in the duodenum of anemic piglets is a decreased expression of both Heph and Fpn, which can be directly caused by changes in the distribution of Cu ions within the enterocyte of anemic piglets and downregulation of Atp7b expression. The obtained results suggest that in anemic piglets, regulation of intra-enterocytic iron absorption is based on the interaction between copper and iron metabolism proteins and dominates systemic demand, even in iron deficiency conditions.

## Figures and Tables

**Figure 1 nutrients-13-00104-f001:**
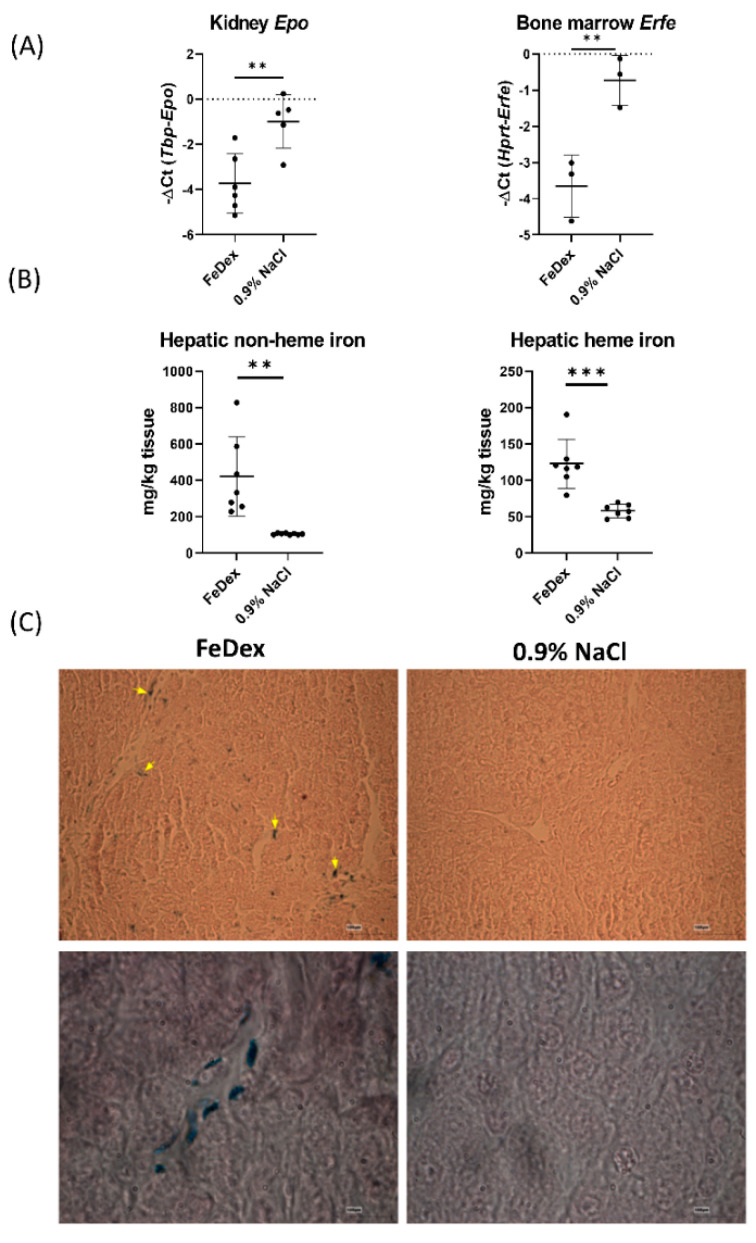
FeDex supplementation improves iron status of piglets. (**A**) mRNA expression level of erythropoietin (Epo) in the kidney and erythroferrone (Erfe) in the bone marrow of 28-day-old anemic (0.9% NaCl) and FeDex supplemented piglets. The results are presented as mean ± SD. Each point reflects the measurement result obtained for a given individual in the group. ** *p* < 0.01. (**B**) The non-heme and heme iron levels in the liver of 28-day-old anemic (0.9% NaCl) and FeDex supplemented piglets. The results are presented as the mean ± SD. ** *p* < 0.01, *** *p* < 0.001. (**C**) The iron deposits in the liver of piglets supplemented with FeDex. The arrow marks observed iron deposits located in liver macrophages. Prussian blue staining (Perls staining) on the 10 µm liver sections, scale bar = 100 µm. Magnification: 20×—upper panel, 100×—lower panel.

**Figure 2 nutrients-13-00104-f002:**
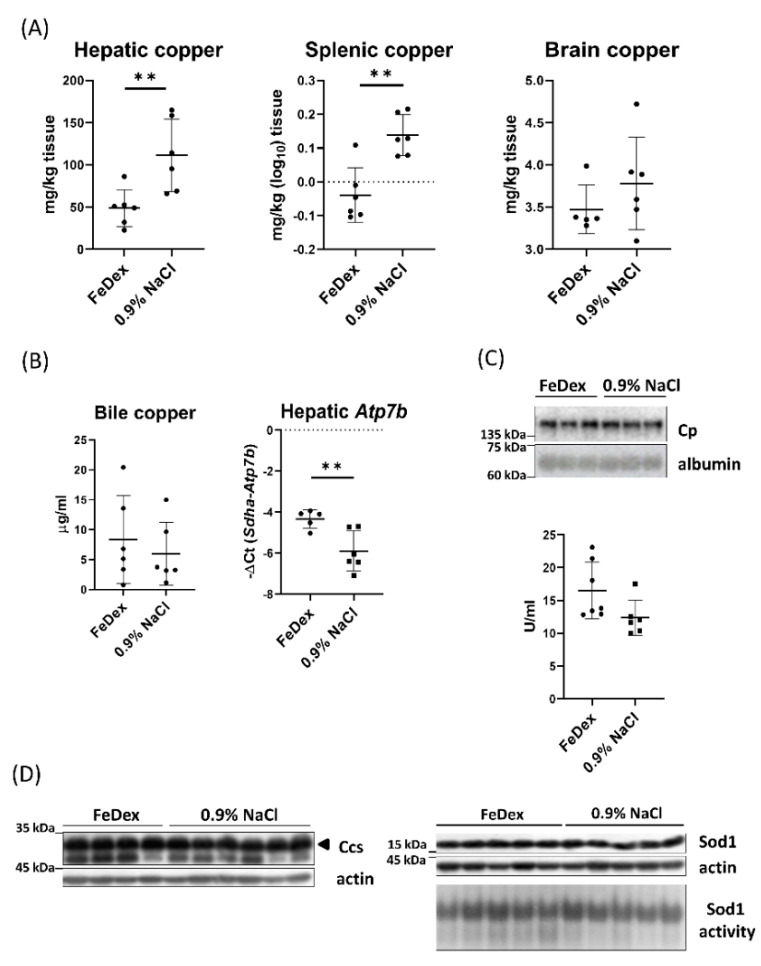
Copper level in the various organs of anemic and FeDex-supplemented piglets. (**A**) The total copper content in the liver, spleen, and brain of 28-day-old anemic (0.9% NaCl) and FeDex-supplemented piglets, measured by the atomic absorption spectrometry (AAS) technique. The graph illustrating the level of copper in the spleen was made based on the statistical transformation (log_10_) of the raw data to gain normal distribution of samples. The results are presented as mean ± standard deviation (SD). Each point reflects the measurement result obtained for a given individual in the group. Number of piglets in group *n* = 6. (**B**) Biliary copper content and Atp7b mRNA expression in the liver of piglets, the results are presented as mean ± SD. (**C**) Western blot analysis of ceruloplasmin protein level in the serum and colorimetric measurement of its activity. (**D**) Western blot analysis of Ccs and Sod1 proteins level in the cytosolic fraction of duodenal scrapings of piglets and in-gel Sod1 activity staining in cytosolic fraction obtained from duodenum. ** *p* < 0.01.

**Figure 3 nutrients-13-00104-f003:**
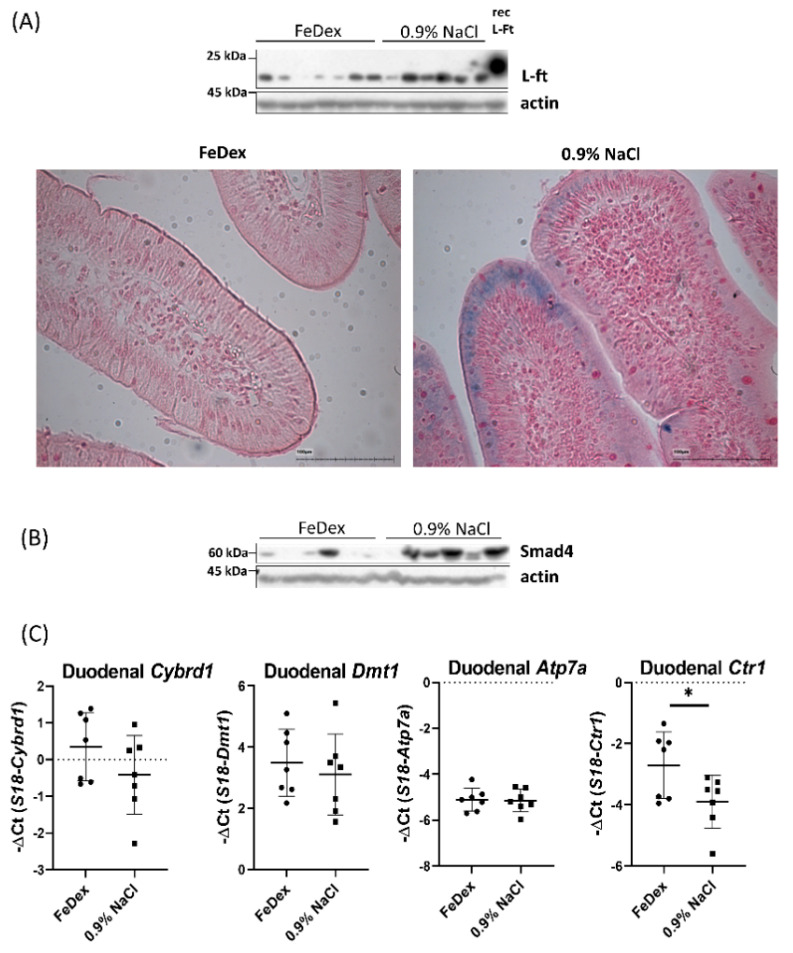
Iron retention in the duodenal enterocytes of anemic piglets. (**A**) Western blot analysis of ferritin light chain (L-Ft) protein level in the cytosolic fraction of duodenal scrapings of piglets. Rec L-Ft—recombinant mouse ferritin light chain was used as a positive control. Perls staining on the 10 µm duodenal sections revealed iron deposition in the duodenal enterocytes of anemic piglets, scale bar = 100 µm. Magnification: 20×. (**B**) Western blot analysis of Smad4 protein level in the total extract of duodenal scrapings. (**C**) The expression of Cybrd1, Dmt1, Atp7a, and Ctr1 at the mRNA level in the duodenum of different groups of piglets. The results are presented as mean ± SD. * *p* < 0.05.

**Figure 4 nutrients-13-00104-f004:**
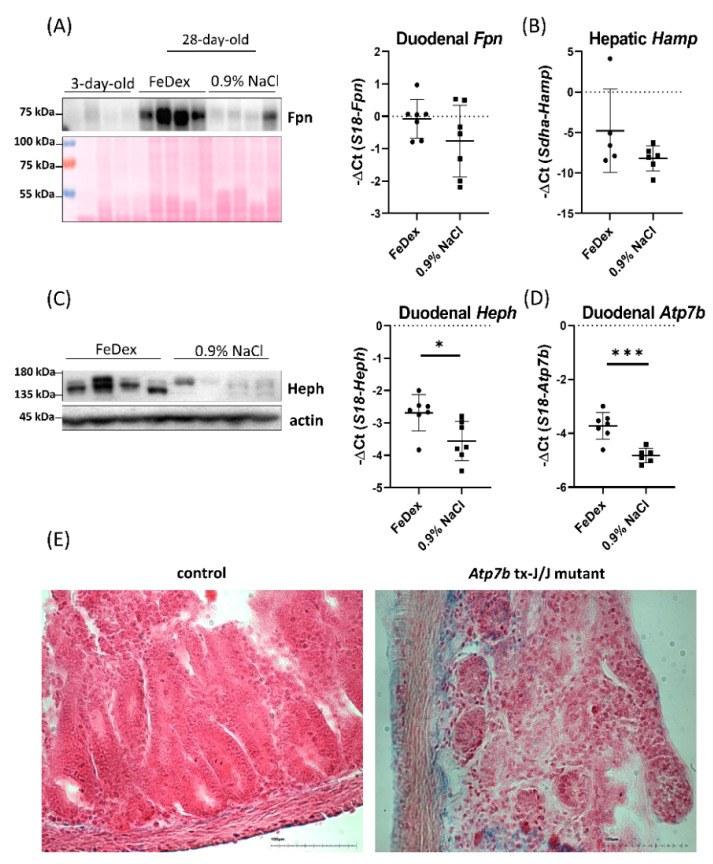
Decreased Atp7b expression affects Heph and Fpn protein levels in the duodenum of anemic piglets. (**A**) Western blot analysis of Fpn protein level in the membrane fraction of duodenal scrapings of three-day-old and 28-day-old piglets with different iron status. The Ponceau-S staining was used as a reference for equal protein loading control. The graph presents the expression of Fpn at the mRNA level in the duodenum of different groups of piglets. The results are presented as mean ± SD. (**B**) The expression of Hamp at the mRNA level in the liver of piglets. (**C**) Western blot analysis of Heph protein level in the membrane fraction of duodenal scrapings and Heph mRNA expression in the duodenum of different groups of piglets. (**D**) The expression of Atp7b at the mRNA level in the duodenum of piglets. * *p* < 0.05, *** *p* < 0.001. (**E**) Perls staining on the 10 µm thick duodenal sections derived from control and Atp7b tx-J/J mice, scale bar = 100 µm. Magnification: 20×.

**Table 1 nutrients-13-00104-t001:** Hematological parameters of piglets.

RBC Indices/Group		RBC (10^6^/mm^3^)	HB (g/dL)	HCT (%)	MCV (µm^3^)
three-day-old		3.44 ± 0.97	6.83 ± 1.66	19.79 ± 5.34	57.67 ± 4.44
28-day-old	0.9% NaCl	3.00 ± 0.92	4.11 ± 0.97 ^‡^*******	10.67 ± 3.05 ^‡^*******	35.89 ± 1.69 ^§^*******
	FeDex	6.13 ± 0.66 ^†^*******	10.73 ± 1.31 ^†^*******	33.41 ± 4.08 ^†^*******	54.56 ± 5.90

RBC—red blood cells; HB—hemoglobin; HCT—hematocrit, MCV—mean corpuscular volume. The basic hematological analysis was performed on the group of three–day–old piglets and two groups of 28-day-old piglets: a group given 0.9% NaCl intramuscularly on the third day of life (0.9% NaCl) and a group of piglets given FeDex intramuscularly, at a dose of 80 mg Fe/kg body weight on the third day of life (FeDex). Number of piglets in each group—*n* = 9. Data expressed as mean ± SD. The symbol ^†^*** means statistical significance at the level *p* < 0.001 between the FeDex group and the other groups. The symbol ^‡^*** means statistical significance at the level of *p* < 0.001 between the 0.9% NaCl group and the group of 3-day-old piglets. The symbol ^§^*** indicates statistical significance at the level *p* < 0.001 between the 0.9% NaCl group and the other groups.

## Data Availability

The authors declare that the data supporting the findings of this study are available within the paper and its Appendix A.

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
