# Peer review of "Relationship between Down-Regulation of Copper-Related Genes and Decreased Ferroportin Protein Level in the Duodenum of Iron-Deficient Piglets"

_nutrients, 2020, doi:10.3390/nu13010104_

Round 1

Reviewer 1 Report

This a quite interesting study which examines iron absorption in piglets and how this is influenced by copper and copper metabolism-related genes/proteins. The topic is of importance given that pigs are valuable agricultural animals and IDA naturally occurs in piglets and iron supplementation is routine. Further, pigs are thought to be an excellent model for human intestinal nutrient absorption. Overall, the paper is well written and most results are clear. There are, however, a few issues that the authors may wish to consider in a revised paper. consideration of these issues raised here could strengthen the paper and potentially increase it spotential impact in the important areas of iron biology and physiology. Specific comments follow below. 

1) The fact that suckling piglets are naturally iron deficient is intriguing. Can the authors expound upon this key point? If iron deficiency naturally occurs in piglets, why should it be corrected? Does the low iron environment provide protection against neonatal infections (for example)? It is clear why piglets are supplemented with iron in the agriculture industry (to increase growth), but what is the significance of low iron in the context of normal pig physiology?

2) Please provide a rationale for using male piglets. Also, a justification for the amount of iron dextran used for supplementation should be provided (it's presumably based upon other studies published in the scientific literature, but this should be explicitly stated).  

3) Several antibodies were used in this study. The authors should describe evidence demonstrating that these antibodies are indeed detecting the correct protein. Evidence could include experiments performed by the authors of this paper, studies done by others, or information on specificity provided by the commercial suppliers of the various antibodies.

4) It's intriguing that Fpn1 protein expression increased with iron supplementation in duodenal enterocyte membrane fractions. What's further intriguing is that this happened in the absence of hepatic Hamp suppression. Can the authors speculate as to how this might work? This is particularly important as liver-derived hepcidin is thought to be the main regulator of intestinal iron absorption (and hepatic hepcidin mRNA expression is thought to correlate with circulating hormone levels).   

5) In Fig. 5C, the immunoreactive Heph proteins looks to be different sizes (masses). Can the authors provide an explanation for this interesting result?

6) The authors purport to show iron staining of liver macrophages (i.e., presumably Kupffer cells) in iron-supplemented piglets. From the image provided, however, it is not clear that the cells pointed out are indeed macrophages. A more thorough approach would have utilized a macrophage-specific marker and shown colicalization for iron deposits. As it stands, the conclusion is thus not strong. 

7) The discussion is very long and somewhat speculative. Can it be shortened and more directly focused on aspects of iron/copper metabolism that are specifically, experimentally addressed in this paper? 

Reviewer 2 Report

The authors  attempted to investigate the influence of copper-related genes on iron absorption in a duodenum of animal model such as piglets  with different iron status. They focused their attention on 3 specific proteins. 

The paper is well written and documented and the study has been carefully conducted.   Minor points should to be addressed:

there are some typing errors

Labels in the figures could be magnified in order to improve reading

some references should be changed (1956, 1988) and substituted with more recent and complete papers

the results paraghraphs are difficult to read. They should be simplified moving comments to the discussion section
